# Risk and symptoms of COVID-19 in health professionals according to baseline immune status and booster vaccination during the Delta and Omicron waves in Switzerland—A multicentre cohort study

Baharak Babouee Flury[1], Sabine Güsewell[1], Thomas Egger[1], Onicio Leal[2,3], Angela Brucher[4], Eva Lemmenmeier[5], Dorette Meier Kleeb[6], J. Carsten Möller[7], Philip Rieder[8], Markus Rütti[9], Hans-Ruedi Schmid[10], Reto Stocker[8], Danielle Vuichard-Gysin[11], Benedikt Wiggli[12], Ulrike Besold[13], Allison McGeer[14], Lorenz Risch[15,16,17], Andrée Friedl[12], Matthias Schlegel[1], Stefan P. Kuster[1], Christian R. Kahlert[1,18�*], Philipp Kohler[1�*], on behalf of the SURPRISE Study Group[¶]

1 Cantonal Hospital St Gallen, Division of Infectious Diseases and Hospital Epidemiology, St Gallen, Switzerland, 2 Epitrack, Recife, Brazil, 3 Department of Economics, University of Zurich, Zurich, Switzerland, 4 Psychiatry Services of the Canton of St. Gallen (South), St Gallen, Switzerland, 5 Clienia Littenheid AG, Private Clinic for Psychiatry and Psychotherapy, Littenheid, Switzerland, 6 Kantonsspital Baden, Division of Occupational Health, Baden, Switzerland, 7 Center for Neurological Rehabilitation, Zihlschlacht, Switzerland, 8 Hirslanden Clinic, Zurich, Switzerland, 9 Fuerstenland Toggenburg Hospital Group, Wil, Switzerland, 10 Kantonsspital Baden, Central Laboratory, Baden, Switzerland, 11 Thurgau Hospital Group, Division of Infectious Diseases and Hospital Epidemiology, Muensterlingen, Switzerland, 12 Kantonsspital Baden, Division of Infectious Diseases and Hospital Epidemiology, Baden, Switzerland, 13 Geriatric Clinic St. Gallen, St. Gallen, Switzerland, 14 Sinai Health System, Toronto, Canada, 15 Labormedizinisches Zentrum Dr Risch Ostschweiz AG, Buchs, Switzerland, 16 Private Universität im Fürstentum Liechtenstein, Triesen, Liechtenstein, 17 Center of Laboratory Medicine, University Institute of Clinical Chemistry, University of Bern, Inselspital, Bern, Switzerland, 18 Children's Hospital of Eastern Switzerland, Department of Infectious Diseases and Hospital Epidemiology, St. Gallen, Switzerland

055 These authors contributed equally to this work.
¶ Membership of the SURPRISE Study Group is provided in the Acknowledgments.
* christian.kahlert@kssg.ch (CRK); philipp.kohler@kssg.ch (PK)

**Data Availability Statement:** Data are available at https://dx.doi.org/10.5281/zenodo.7149075.

## Abstract

### Background

Knowledge about protection conferred by previous Severe Acute Respiratory Syndrome Coronavirus 2 (SARS-CoV-2) infection and/or vaccination against emerging viral variants allows clinicians, epidemiologists, and health authorities to predict and reduce the future Coronavirus Disease 2019 (COVID-19) burden. We investigated the risk and symptoms of SARS-CoV-2 (re)infection and vaccine breakthrough infection during the Delta and Omicron waves, depending on baseline immune status and subsequent vaccinations.

### Methods and findings

In this prospective, multicentre cohort performed between August 2020 and March 2022, we recruited hospital employees from ten acute/nonacute healthcare networks in Eastern/

**Funding:** This work was supported by the Swiss National Sciences Foundation (grant number 31CA30_196544 to PK and CRK; grant number PZ00P3_179919 to PK), the Federal Office of Public Health (grant number 20.008218/421-28/1), and the Health Department of the Canton of St. Gallen. Funders played no role in the study design, data collection and analysis, decision to publish, or preparation of the manuscript.

**Competing interests:** The authors have declared that no competing interests exist.

**Abbreviations:** aHR, adjusted HR; anti-N, anti-nucleocapsid; anti-S, anti-spike; aRR, adjusted rate ratio; CI, confidence interval; COVID-19, Coronavirus Disease 2019; HCW, healthcare worker; HR, hazard ratio; NPS, nasopharyngeal swab; PCR, polymerase chain reaction; RAD, rapid antigen diagnostic; RR, risk ratio; SARS-CoV-2, Severe Acute Respiratory Syndrome Coronavirus 2.

Northern Switzerland. We determined immune status in September 2021 based on serology and previous SARS-CoV-2 infections/vaccinations: Group N (no immunity); Group V (twice vaccinated, uninfected); Group I (infected, unvaccinated); Group H (hybrid: infected and $\geq 1$ vaccination). Date and symptoms of (re)infections and subsequent (booster) vaccinations were recorded until March 2022. We compared the time to positive SARS-CoV-2 swab and number of symptoms according to immune status, viral variant (i.e., Delta-dominant before December 27, 2021; Omicron-dominant on/after this date), and subsequent vaccinations, adjusting for exposure/behavior variables.

Among 2,595 participants (median follow-up 171 days), we observed 764 (29%) (re) infections, thereof 591 during the Omicron period. Compared to group N, the hazard ratio (HR) for (re)infection was 0.33 (95% confidence interval [CI] 0.22 to 0.50, $p < 0.001$) for V, 0.25 (95% CI 0.11 to 0.57, $p = 0.001$) for I, and 0.04 (95% CI 0.02 to 0.10, $p < 0.001$) for H in the Delta period. HRs substantially increased during the Omicron period for all groups; in multivariable analyses, only belonging to group H was associated with protection (adjusted HR [aHR] 0.52, 95% CI 0.35 to 0.77, $p = 0.001$); booster vaccination was associated with reduction of breakthrough infection risk in groups V (aHR 0.68, 95% CI 0.54 to 0.85, $p = 0.001$) and H (aHR 0.67, 95% CI 0.45 to 1.00, $p = 0.048$), largely observed in the early Omicron period. Group H (versus N, risk ratio (RR) 0.80, 95% CI 0.66 to 0.97, $p = 0.021$) and participants with booster vaccination (versus nonboosted, RR 0.79, 95% CI 0.71 to 0.88, $p < 0.001$) reported less symptoms during infection. Important limitations are that SARS-CoV-2 swab results were self-reported and that results on viral variants were inferred from the predominating strain circulating in the community at that time, rather than sequencing.

## Conclusions

Our data suggest that hybrid immunity and booster vaccination are associated with a reduced risk and reduced symptom number of SARS-CoV-2 infection during Delta- and Omicron-dominant periods. For previously noninfected individuals, booster vaccination might reduce the risk of symptomatic Omicron infection, although this benefit seems to wane over time.

---

## Author summary

### Why was this study done?

- Preexisting immunity against Severe Acute Respiratory Syndrome Coronavirus 2 (SARS-CoV-2)—either from previous infection or vaccination—confers protection against severe Coronavirus Disease 2019 (COVID-19).

- Few studies have prospectively determined the SARS-CoV-2 infection risk based on large-scale serologic testing to detect previous asymptomatic infections.

- Real-world evaluations of infections during periods with distinct SARS-CoV-2 variants are valuable to assess protection by preexisting immunity and inform health policy guidelines.

## What did the researchers do and find?

- In September 2021, 2,554 healthcare workers were classified into four different groups based on previous SARS-CoV-2 serology results and infection/vaccination history.

- Participants were followed until March 2022 to assess the association of immune status and additional vaccinations with self-reported COVID-19 and symptoms.

- Hybrid immunity (i.e., previous infection and at least one vaccination) resulted in a reduced SARS-CoV-2 infection risk and less symptoms during the Delta or Omicron period.

- Booster vaccination was associated with reduced infection risk and less symptoms during the first half of the Omicron period analysed in our study.

## What do these findings mean?

- Individuals who were previously infected and vaccinated seem to be best protected and exhibit less symptoms of SARS-CoV-2 infection than those with other immune status.

- Booster vaccination might further reduce both the risk of Omicron breakthrough infection and the number of reported symptoms, although this benefit fades over time.

- These findings might inform healthcare providers and public health authorities in estimating the risk of SARS-CoV-2 (re)infection in individuals or communities.

## Background

Mitigation of Coronavirus Disease 2019 (COVID-19) relies on establishing an ideally long-lasting immune barrier against Severe Acute Respiratory Syndrome Coronavirus 2 (SARS-CoV-2). Studies from the pre-Omicron era show that specific neutralizing antibodies as marker for humoral immunity reduce the risk of symptomatic (re)infection and thus disease recurrence upon reexposure to a homologous viral strain [1]. Although natural infection with SARS-CoV-2 elicits a broader humoral response than mRNA vaccine administration, levels of neutralizing antibodies are lower after natural infection [2]. Concerning (re)infection however, previous infection is associated with 85% protection over at least 9 months [3]; also, infected individuals might be even better protected than only vaccinated individuals [4]. A still lower reinfection risk has been reported for individuals with hybrid immunity, i.e., SARS-CoV-2 infection plus at least one vaccination [5–10].

Yet, weak or waning humoral response in tandem with the emergence of new viral variants, capable of potentially escaping immune response, lead to a continued risk of reinfection with heterologous SARS-CoV-2 strains [1,11,12]. This is particularly true for the Omicron variant [13], which is the predominant viral strain globally as of October 2022 [14], causing mostly milder infections compared to preceding strains [15–17]. Available evidence from adult and pediatric populations points towards good effectiveness of prior mRNA vaccination against severe COVID-19 and hospitalisation due to Omicron variants, but less so against asymptomatic and symptomatic, mild breakthrough infections [11,18–21]. Similar to the pre-Omicron

era, some studies have shown that previous infection or hybrid immunity (compared to vaccination alone) might provide better protection against the Omicron variant. However, these studies were either small [22] or relied on population-based data [23], ignoring the fact that a substantial proportion of SARS-CoV-2 infections remain undetected with the risk of misclassification [24]. Also, behavior and exposure variables, which are likely to be different between previously vaccinated and unvaccinated individuals, were not considered in these studies. The influence of type and timing of SARS-CoV-2 vaccination on eventual risk of breakthrough infection has been described for previous variants, but not for Omicron [7,25].

Within a prospective, multicentre healthcare worker (HCW) cohort study, we aimed to determine the risk for and symptoms of COVID-19 by comparing Delta- and Omicron-dominant periods, depending on previous immune status based on infection/vaccination history and serology results. Furthermore, we examined the role of booster vaccination on these outcomes. For those with two vaccinations, we assessed the role of mRNA vaccine type and timing of vaccine doses on breakthrough infection risk.

## Methods

### Study design and population

This prospective cohort (SURPRISE) was initiated in summer 2020, after the first COVID-19 wave in Switzerland. The study was approved by the ethics committee of Eastern Switzerland (#2020–00502), and participants provided digital written informed consent. Health professionals (with and without patient contact) aged 16 years or older from ten healthcare networks located in Northern and Eastern Switzerland were included between June 2020 and March 2021. From their inclusion until September 2021, participants were prospectively followed through weekly questionnaires on SARS-CoV-2 infections/vaccinations, and periodic SARS-CoV-2 serology measurements in August 2020, January 2021, and August/September 2021 [26]. Participants without available serology from August/September 2021 were excluded. For the present work, no prespecified analysis plan was designed.

Baseline immune status was assessed as of September 20, 2021 (**Fig 1**), based on previous infection/vaccination history and all available serology results. During the local emergence of the Delta (October to December 2021) and Omicron B.1.1.529.1 (Nextstrain 21K; BA.1) variant (January to March 2022), the follow-up survey through questionnaires was continued at monthly intervals. Information collected included SARS-CoV-2 exposures, reports of nasopharyngeal swab (NPS) tests (positive and negative), symptoms associated with positive NPS, and receipt of subsequent (booster or first) vaccinations.

### SARS-CoV-2 diagnostics

Participants were asked to get tested for SARS-CoV-2 in case of compatible symptoms, according to national recommendations. SARS-CoV-2 was detected by polymerase chain reaction (PCR) or rapid antigen diagnostic (RAD) test, depending on the participating institutions. Some facilities also switched from PCR to RAD in the course. No sequencing for determination of the viral variants was performed; the viral variant was inferred from the predominating strain circulating in the community at that time (see below). To verify the completeness and accuracy of self-reported NPS results (PCR or RAD), all positives and a random sample of negatives were validated for a subgroup of HCWs from the largest participating institution as described previously [27]. Anti-nucleocapsid (anti-N) and anti-spike (anti-S) antibodies were measured using the Roche Elecsys (Roche Diagnostics, Rotkreuz, Switzerland) electro-chemi-luminescence immunoassay [28].

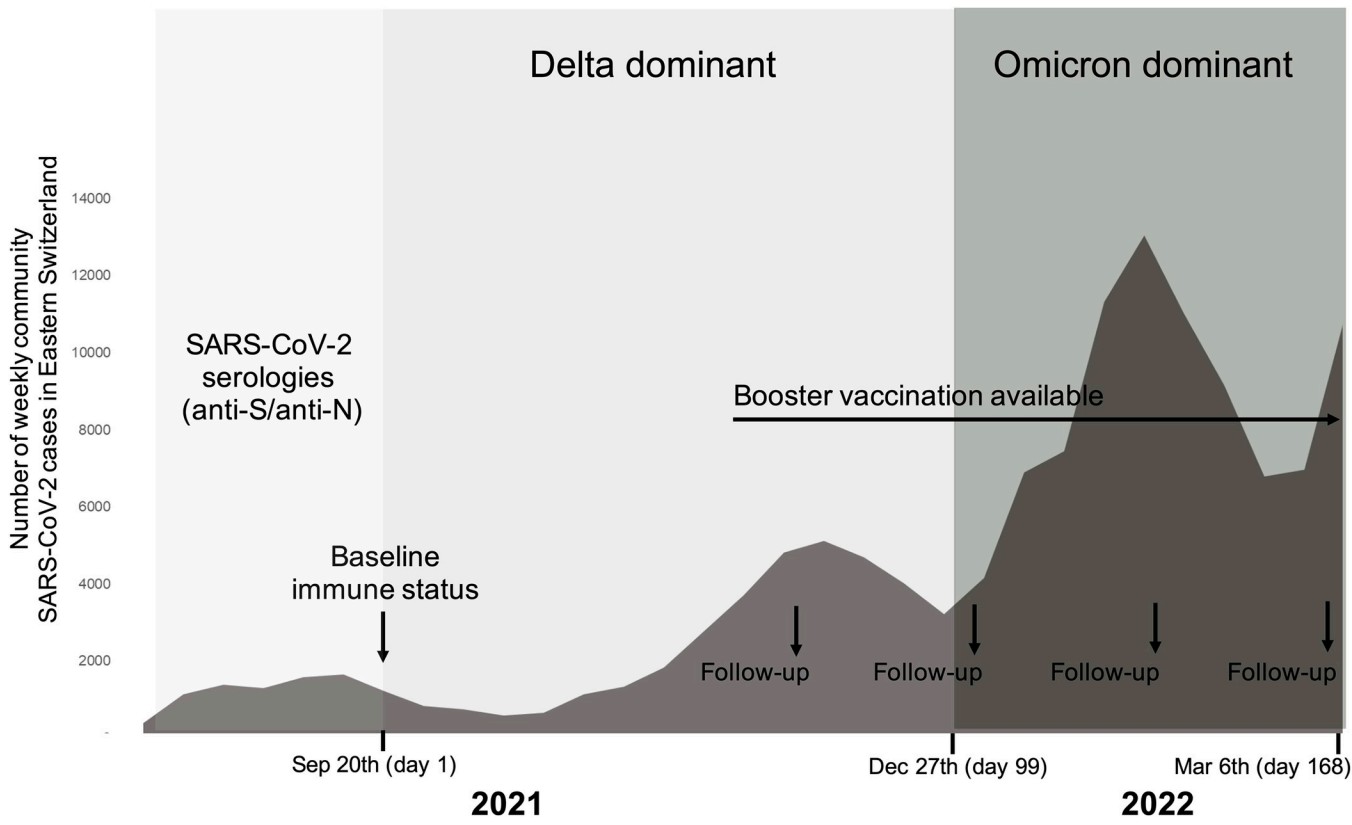

**Fig 1. Study timeline depicting the longitudinal follow-up of participants in correlation with the virus variants circulating in Eastern Switzerland.**

### Definition of predictor variables

We defined four distinct immune status groups (i.e., main predictor) as of September 20, 2021, according to previous questionnaires and serology results: (i) Group N (reference): no reported infection and anti-N/anti-S negative and no previous SARS-CoV-2 vaccination; (ii) Group V (vaccinated): no reported infection and anti-N negative, but twice vaccinated (with any time interval between doses) with the second dose being at least 7 days ago; (iii) Group I (infected): infection reported or anti-N positive (at any time), but no vaccination; (iv) Group H (hybrid immunity): reported infection or anti-N positive (at any time) and vaccination ($\geq$1 dose) at least 7 days ago. In addition, we collected information on vaccination after September 20, 2021, either booster vaccination (available from November 2021 on) for groups V and H or first vaccinations for groups N and I. Again, participants were considered as boosted 7 days after receipt of the vaccine. Immune status was treated as time-dependent variable, so that participants switched from group I to H after the first vaccination, and from group N to V after the second vaccination. Participants in group N with only one vaccination were not considered and were only included 7 days after receipt of their second dose. The different groups and outcomes are illustrated in **S1 Fig**. For other predictor variables, **S1 Table** shows definitions and time points of the corresponding questionnaire.

### Outcomes

The main outcome was time to the first SARS-CoV-2-positive NPS reported after September 20, 2021. Outcomes occurring between September 20, 2021 and March 6, 2022 were included.

The period before December 27, 2021 was defined as Delta-dominant (i.e., Delta period), the period on and after this date as Omicron-dominant (i.e., Omicron period), based on sequencing data from North-Eastern Switzerland [29]. We treated our outcome as a survival event, i.e., participants were no longer at risk after their first positive NPS. We also calculated the number of symptoms reported during SARS-CoV-2 infection.

## Statistical analysis regarding infection risk

We computed Kaplan–Meier curves to compare the occurrence of SARS-CoV-2 (re)infection (first positive NPS) through time according to immune status; noninfected participants were censored at the time of the last available follow-up questionnaire. The risk of (re)infection was compared among groups (treated as time-dependent variable) using Cox regression; hazard ratios (HRs) and corresponding 95% confidence intervals (CIs) were calculated. This analysis was performed separately for the Delta and Omicron periods, with and without adjusting for receipt of booster vaccination. In addition, Kaplan–Meier curves were computed to visualize the association of booster vaccination on infection risk during the Omicron period, when most of these vaccinations had already been received.

Multivariable Cox-regression analysis was performed to correct for additional confounding variables, which were a priori selected based on previous analyses and expected importance [30]. Multiple imputation was used to substitute missing values (**S1 Methods**). Time-dependent variables were, besides immune status, receipt of booster vaccination, SARS-CoV-2 infection of a household contact within the same month, and documentation of ≥1 negative test in the previous month (to adjust for differences in testing behaviour). Time-independent variables included baseline anthropometric data and further variables reflecting SARS-CoV-2 risk exposures and behaviours (**S1 Table**). We performed a sensitivity analysis excluding infections occurring during the period of variant overlap between December 6, 2021 and January 3, 2022 to minimize contamination of infections occurring in the Delta and Omicron periods with the respective other viral variant.

## Supplementary analyses

Because model diagnostics showed that the influence of booster vaccination was time dependent in the Omicron period, we further split this period into an early (before February 15, 2022) and late Omicron phase (after this date).

To estimate the influence of time since last immunization event, we included time from previous infection or vaccination (i.e., time of preimmunization) until September 20, 2021 for those individuals for which this information was available. The exact day of initial infection could not be assessed in those with only positive anti-N but no report of positive NPS; also, group N, which per definition did not have an immunization event, was excluded and group V was chosen as reference category instead.

Finally, to assess the impact of type and timing of vaccination within group V, we included the type of vaccine (i.e., mRNA-1273 or BNT162b2), the time of preimmunization, and time between first and second vaccination dose.

## Frequency of SARS-CoV-2 symptoms

We used univariable and multivariable Poisson regression to assess the impact of baseline immune status on symptom number for infections that occurred before any booster or first vaccination. Covariables included the virus variants (Delta versus Omicron period), the month of (re)infection (to adjust for the fact that immune protection wanes over time), and a priori selected variables based on their importance in previous analyses [31].

To assess the impact of booster vaccination on number of symptoms, we performed a second analysis restricted to groups V and H, and to infections occurring during the Omicron period (because of the small number of infections preceded by booster in the Delta period). The study is reported as per the Strengthening the Reporting of Observational Studies in Epidemiology (STROBE) guideline (**S1 STROBE Checklist**) [32]. R version 4.1.2 was used for statistical analyses.

## Results

### Study population

Of the 5,792 initial cohort participants, 2,595 (45%) underwent serology testing in August/September 2021 and completed at least one follow-up questionnaire. Thereof, 2,554 were classified into one of the immune status groups at baseline: 581 (22.7%) in group H, 162 (6.3%) in I, 1,643 (64.3%) in V, and 168 (6.6%) in N; additionally, 41 individuals were assigned to group V during follow-up (after their second vaccination) (**S2 Fig**). Median follow-up was 171 days (interquartile range 131 to 171). Baseline characteristics are summarized in **Table 1**.

### SARS-CoV-2 (re)infections by immune status and time period

A total of 764 (29.4%) infections were reported in 2,595 participants, whereof 173 (22.6%) occurred during the Delta and 591 (77.4%) during the Omicron period. During Delta, the risk for COVID-19 was significantly reduced for groups V (HR 0.33, 95% CI 0.22 to 0.50, $p < 0.001$), I (HR 0.25, 95% CI 0.11 to 0.57, $p = 0.001$), and H (HR 0.04, 95% CI 0.02 to 0.10, $p < 0.001$) compared to group N. These associations were less pronounced during the Omicron period for all groups (**Fig 2**).

### Booster vaccination and risk of infection

By March 2022, 80% of participants in groups V and H had received a booster (median time of booster was December 14, 2021) (**S3 Fig**). Adjusting the univariable model for booster vaccination, the adjusted HR (aHR) remained similar for the Delta period as in the unadjusted

**Table 1. Baseline characteristics of study participants ($n = 2,582$) according to immune status at baseline.** Number ($n$) of individuals with missing values are also indicated.

| Immune status at baseline | N | V | I | H | Missing |
|---|---|---|---|---|---|
| | $n = 168$ | $n = 1,643$ | $n = 162$ | $n = 581$ | ($n$) |
| Age at baseline (median, years) | 38.3 | 45.2 | 35.2 | 41.7 | 4 |
| Sex (% male vs. female) | 8.4% | 21.5% | 8.7% | 20.7% | 22 |
| Body mass index >30 kg/m$^2$ at study inclusion (%) | 10.1% | 12.1% | 12.3% | 11.2% | 0 |
| Any comorbidity at study inclusion (%) | 31.1% | 40.1% | 32.1% | 40.4% | 120 |
| Any patient contact at baseline (%) | 79.1% | 76.5% | 87.9% | 86.3% | 117 |
| Always wearing respirator mask at baseline (%) | 13.9% | 22.0% | 9.4% | 18.8% | 117 |
| Anti-spike titer (median, in BAU/ml) | <0.4 | 1085 | 103 | 3502 | 82 |
| Any negative test during follow-up (%) | 83.3% | 73.4% | 74.1% | 71.6% | 0 |
| Any positive household member during follow-up (%) | 36.9% | 34.8% | 28.4% | 26.1% | 0 |
| Duration of follow-up (median, days) | 171 | 171 | 171 | 171 | 0 |

None (N): no reported infection and anti-N/-S negative and no previous SARS-CoV-2 vaccination; V (vaccinated): no reported infection and anti-N negative, but twice vaccinated; I (infected): infection reported or anti-N positive (at any time), but no vaccination; H (hybrid immunity): reported infection or anti-N positive (at any time) and vaccination (≥1 dose).

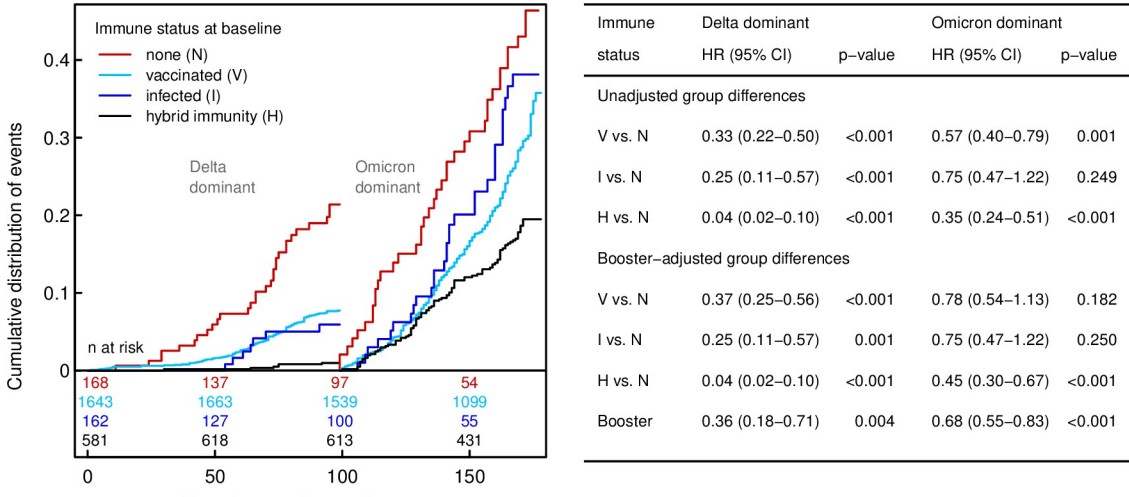

| Immune status | Delta dominant | | Omicron dominant | |
|---|---|---|---|---|
| | HR (95% CI) | p–value | HR (95% CI) | p–value |
| Unadjusted group differences | | | | |
| V vs. N | 0.33 (0.22–0.50) | <0.001 | 0.57 (0.40–0.79) | 0.001 |
| I vs. N | 0.25 (0.11–0.57) | <0.001 | 0.75 (0.47–1.22) | 0.249 |
| H vs. N | 0.04 (0.02–0.10) | <0.001 | 0.35 (0.24–0.51) | <0.001 |
| Booster–adjusted group differences | | | | |
| V vs. N | 0.37 (0.25–0.56) | <0.001 | 0.78 (0.54–1.13) | 0.182 |
| I vs. N | 0.25 (0.11–0.57) | 0.001 | 0.75 (0.47–1.22) | 0.250 |
| H vs. N | 0.04 (0.02–0.10) | <0.001 | 0.45 (0.30–0.67) | <0.001 |
| Booster | 0.36 (0.18–0.71) | 0.004 | 0.68 (0.55–0.83) | <0.001 |

**Fig 2.** Left: influence of baseline immune status on the time course of (re)infection events, shown separately for the two periods (Delta vs. Omicron) by resetting Kaplan–Meier curves to 0 on December 27, 2021. Note that group differences depicted in the graph include any impact of booster vaccination (in groups V and H). Right: HRs with 95% CIs from Cox regression regarding risk of SARS-CoV-2 (re)infection for each immune status compared with group N (no previous infection or vaccination), both without and with adjustment for booster vaccination. CI, confidence interval; HR, hazard ratio; SARS-CoV-2, Severe Acute Respiratory Syndrome Coronavirus 2.

model. In the Omicron period, only group H showed a reduced risk (aHR 0.45, 0.30 to 0.67, $p < 0.001$) compared to group N, whereas no significant risk reduction was observed for groups V and I (**Fig 2**). Restricting the analysis to the Omicron period, receipt of booster vaccination was associated with reduced infection risk in groups V (aHR 0.68, 95% CI 0.54 to 0.85, $p = 0.001$) and H (aHR 0.67, 95% CI 0.45 to 1.00, $p = 0.048$) (**Fig 3**).

## Multivariable analysis and sensitivity analyses

When models included exposure and behaviour variables, results were similar, with group H being the only group with reduced infection risk (aHR 0.52, 95% CI 0.35 to 0.77, $p = 0.001$) in

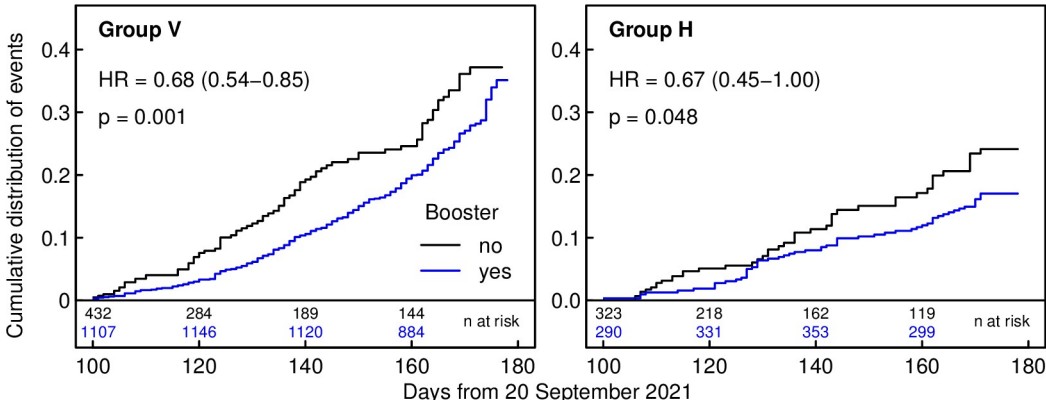

**Fig 3. Influence of booster on the time course of (re)infection events during Omicron dominance, with HRs and 95% CIs, according to immune status.** Note that participants receiving their booster after December 27, 2021 were initially classified as "no" and subsequently switched to "yes," so that numbers at risk for "yes" increase initially. CI, confidence interval; HR, hazard ratio.

**Table 2. HRs with 95% CIs from multivariable Cox regression regarding COVID-19 risk according to immune status and period.**

| | Delta period (*n* = 2,593) | | Omicron period (*n* = 2,355) | |
|---|---|---|---|---|
| | aHR (95% CI) | *p*-value | aHR (95% CI) | *p*-value |
| Group V (vs. N) | 0.47 (0.31–0.74) | 0.001 | 0.85 (0.58–1.23) | 0.393 |
| Group I (vs. N) | 0.26 (0.11–0.59) | 0.002 | 0.74 (0.45–1.19) | 0.213 |
| Group H (vs. N) | 0.06 (0.02–0.14) | <0.001 | 0.52 (0.35–0.77) | 0.001 |
| Age (per decade) | 0.97 (0.84–1.13) | 0.709 | 0.79 (0.73–0.86) | <0.001 |
| Male vs. female | 1.16 (0.80–1.69) | 0.439 | 0.87 (0.70–1.08) | 0.193 |
| Body mass index >30 kg/m$^2$ | 0.84 (0.50–1.41) | 0.518 | 1.02 (0.79–1.31) | 0.890 |
| Patient contact | 0.77 (0.53–1.11) | 0.167 | 0.89 (0.72–1.09) | 0.253 |
| Respirator mask use | 0.99 (0.64–1.55) | 0.980 | 1.09 (0.87–1.35) | 0.461 |
| Positive household in last month | 9.66 (7.15–13.05) | <0.001 | 6.17 (5.24–7.27) | <0.001 |
| Any negative test | 1.26 (0.90–1.75) | 0.181 | 1.11 (0.93–1.32) | 0.236 |
| Booster | 0.42 (0.21–0.83) | 0.014 | 0.81 (0.66–0.99) | 0.043 |

N (no immunity): no reported infection and anti-N/-S negative and no previous SARS-CoV-2 vaccination; V (vaccinated): no reported infection and anti-N negative, but twice vaccinated; I (infected): infection reported or anti-N positive (at any time), but no vaccination; H (hybrid immunity): reported infection or anti-N positive (at any time) and vaccination (≥1 dose).

aHR, adjusted HR; CI, confidence interval; statistics obtained by pooling model coefficients and standard errors from ten imputations of missing covariate values; COVID-19, Coronavirus Disease 2019; HR, hazard ratio; SARS-CoV-2, Severe Acute Respiratory Syndrome Coronavirus 2.

For results of univariable analysis, see S2 Table.

the Omicron period (Table 2). The main risk correlate for SARS-CoV-2 positivity was having a SARS-CoV-2-positive household (as reported by the participant), both in the Delta (aHR 9.66, 95% CI 7.15 to 13.05, *p* < 0.001) and the Omicron period (aHR 6.17, 95% CI 5.24 to 7.27, *p* < 0.001).

Excluding infections during the Delta-Omicron overlap period and performing further missing value imputations as well as a complete case analysis yielded similar HRs as the main analysis (S3 Table).

## Supplementary analyses

Splitting the Omicron period into an early and late phase suggested a benefit of the booster vaccination for the early Omicron phase (aHR 0.60, 95% CI 0.47 to 0.76, *p* < 0.001), but not for the late Omicron phase (aHR 1.02, 95% CI 0.75 to 1.38, *p* = 0.90) (S4 Table).

Time since preimmunization was associated with infection risk during the Delta period (aHR 1.16 per additional months, 95% CI 1.06 to 1.28, *p* = 0.003), but not the Omicron period (aHR 1.00, 95% CI 0.96 to 1.04, *p* = 0.82). Hybrid immunity remained the immune status, which best protected against infection in both periods. Of note, group I (compared to group V) was better protected against infection in the Delta period (aHR 0.27, 95% CI 0.09 to 0.82, *p* = 0.023), but not in the Omicron period (aHR 0.84, 95% CI 0.49 to 1.44, *p* = 0.527) (S5 Table).

Within group V, neither the time interval between dose 1 and dose 2 (HR 0.96 per month, 95% CI 0.73 to 1.27 *p* = 0.788) nor receipt of the mRNA-1273 (versus BNT162b2) vaccine (HR 0.79, 95% CI 0.62 to 1.01, *p* = 0.060) were significantly associated with reduced infection risk (S6 Table).

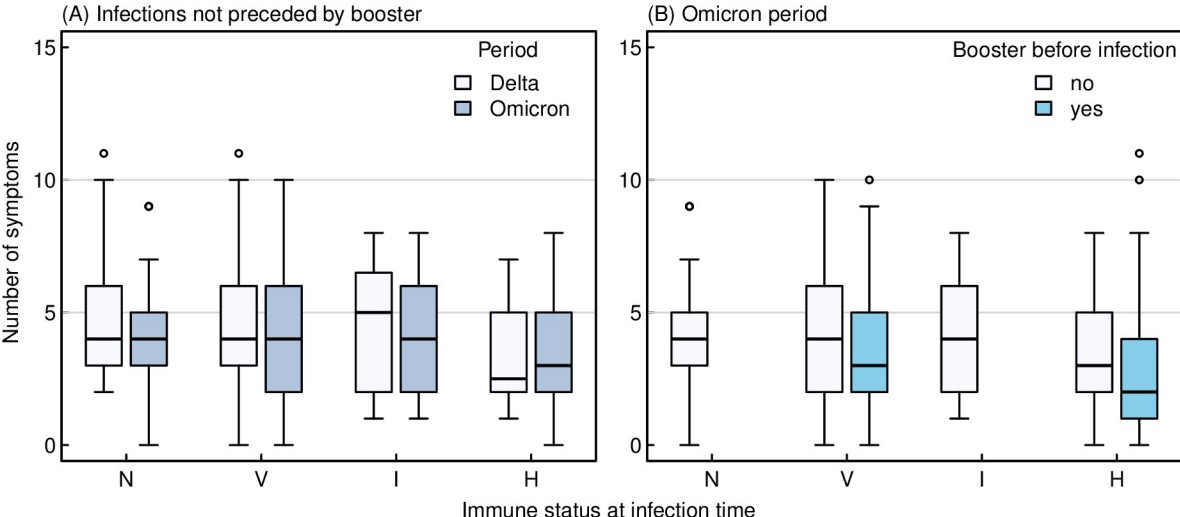

**Fig 4.** Number of symptoms reported after (re)infection by baseline immune status, grouped by Delta and Omicron periods (panel A, left) and receipt of booster (panel B, right). N (no immunity): no reported infection and anti-N/-S negative and no previous SARS-CoV-2 vaccination; V (vaccinated): no reported infection and anti-N negative, but twice vaccinated; I (infected): infection reported or anti-N positive (at any time), but no vaccination; H (hybrid immunity): reported infection or anti-N positive (at any time) and vaccination (≥1 dose).

## SARS-CoV-2 symptoms according to immune status and time period

Participants reported a median of four symptoms during SARS-CoV-2 episodes in the Delta period, and three in the Omicron period (**Fig 4**). Group H reported fewer symptoms (median of three) than group N (median of four, adjusted rate ratio [aRR] 0.81, 95% CI 0.67 to 0.97, $p = 0.026$), even after adjusting for multiple covariables. Groups V and I did not report less symptoms compared to group N (**S7 Table**). Restricting the analysis to vaccinated participants, hybrid immunity (group H versus V, aRR 0.80, 95% CI 0.71 to 0.91, $p < 0.001$) and receipt of booster vaccination (aRR 0.79, 95% CI 0.71 to 0.88, $p < 0.001$), were associated with fewer symptoms, whereas having a comorbidity at baseline (aRR 1.17, 95% CI 1.06 to 1.28, $p = 0.001$) and being infected later during the Omicron period (aRR 1.22 per additional months, 95% CI 1.13 to 1.31, $p < 0.001$) were associated with more symptoms (**S8 Table**).

## Discussion

In this prospective multicentre study, we observed that participants with hybrid immunity and those receiving booster vaccination reported the lowest risk for COVID-19 during the Delta and Omicron periods, and, when infected during the Omicron period, were less symptomatic than those only vaccinated or previously infected. However, this association with booster vaccination waned over time. Type and timing of baseline mRNA vaccines were not associated with the outcomes.

Hybrid immunity, defined as immunity acquired from previous infection plus at least one vaccination, was shown to be associated with reduced risk of SARS-CoV-2 reinfection compared to previous infection only, for up to 9 months [33]. We confirm findings of this Swedish study, which ended in October 2021 (i.e., before Delta was the predominant variant in Europe including Sweden) [34]. Also, our data are in line with a study from Israel performed mainly during the Delta period showing that persons with hybrid immunity were better protected against breakthrough infection compared to only vaccinated persons [10]. In addition, our findings suggest that hybrid immunity acquired from infections with previous variants not

only provides protection against infections by the Delta, but also the Omicron variant, as described in a previous study from Qatar [23]. In contrast to this latter study, which relied on population-level data, we used baseline serology results allowing us to additionally capture previous asymptomatic infections and to adequately assign participants to the respective immune status groups. Furthermore, our study revealed that hybrid immunity was the only immune status associated with less symptoms compared to the group without any preexisting immunity, for both time periods. These findings are in line with results of a laboratory-based study, which showed that antibodies from sera of people with hybrid immunity were able to better neutralize the Omicron variant, compared to antibodies from only vaccinated or infected individuals [35].

When ignoring the booster, we did no longer observe any additional protection of two-dose vaccination in noninfected participants during the Omicron period. This finding adds to data from Qatar, where the effectiveness of two-dose BNT162b2 vaccination against symptomatic Omicron infection was found to be negligible [23]. For the Delta period, we observed a higher risk of (re)infection in twice vaccinated compared to previously infected, nonvaccinated participants. Similarly, in a large retrospective observational study in 124,500 participants, naturally acquired immunity to SARS-CoV-2 conferred stronger protection against infection and symptomatic disease caused by the Delta variant, compared to the BNT162b2 two-dose vaccine-induced immunity [36]. The benefit of previous infection could be explained by the broader immune response elicited by natural infection, with humoral and cellular immune responses not only targeting the spike protein but also other viral antigens. However, in our study, this association was no longer apparent during the Omicron period, where the infection risk of participants with only infection-induced immunity compared to those with vaccine-induced immunity was similar.

The effectiveness of previous infection in preventing reinfection with the Alpha, Beta, and Delta variants of SARS-CoV-2 was around 90% and 60% for Omicron in another study from Qatar [37]. These estimates are higher than in our study, where previous infection was associated with a 75% risk reduction during the Delta, and 25% during the Omicron dominating period. In contrast to Altarawneh and colleagues, we included individuals with previous asymptomatic infection. As the humoral immune response elicited by asymptomatic is weaker compared to symptomatic infection [38], the benefit against reinfections might be lower, which could explain the observed discrepancy.

Previous data have shown that the effectiveness of mRNA-1273 might be superior to BNT162b2, likely because of the slower rise and faster decay of neutralizing antibody titers elicited by BNT162b2 [39,40]. Our study was not designed to assess true vaccine effectiveness. However, when adjusting for receipt of booster vaccine, the risk for breakthrough infection in participants receiving either vaccine was similar, as has also been reported from Qatar [23]; the time interval between first and second vaccine did not impact these results, as has shown previously for the ChAdOx1 vaccine [25].

Our data point towards a benefit of booster vaccination against infection in the Omicron period of approximately 30%, which is below the 47% booster effectiveness reported previously [6]. Also, in a study among US adults with COVID-like illness, the odds ratio for boosted versus nonboosted individuals was 0.16 for Delta and 0.34 for Omicron infections [19], which is in the range of a prospective cohort of frontline workers, where booster mRNA vaccine provided around 90% protection against Delta and 60% against Omicron infection [18]. In the latter study, participants were routinely tested for SARS-CoV-2, resulting in a relevant proportion of asymptomatic infections, which could have overestimated the benefit of the booster vaccination. Another explanation for the lower figures observed in our study is that we collected data over a longer period of Omicron activity (over 2 months), during which the

booster effect might have waned [41]. Indeed, most participants in our cohort received their booster vaccination before onset of the Omicron period; at the same time, in a supplementary analysis of our data, the benefit associated with booster vaccination vanished in the late Omicron period.

Nevertheless, both with the Delta and the Omicron variants, mRNA booster lead to strong protection against COVID-19–related hospitalization and death [20,42]. Although these outcomes were not relevant in our context, we observed a reduction of symptoms among boosted versus nonboosted individuals during the Omicron period. However, we did not assess whether this reduction of symptoms led to less work absenteeism or visits to healthcare providers.

The most important strengths of this study are the availability of baseline serology data along with behavioural and exposure variables, the prospective nature, and the coverage of symptoms associated with COVID-19. Limitations of our study include that SARS-CoV-2 testing was not mandatory and that results were self-reported. However, we previously showed that self-reported NPS results were highly consistent with documented (for positive NPS) and nondocumented (negative NPS) infections [27]. In addition, we adjusted our analysis for the participants' testing behaviour. Viral variants were categorized based on the community epidemiology only (predominating viral strain) and not on sequencing results. This potential imprecision could have led to underestimation of the differential impact of viral variant; yet, excluding infections occurring during the overlap period as sensitivity analysis did not significantly change the results. SARS-CoV-2-specific T-cell immunity is also likely relevant, for assessing the risk for (re)infection, particularly of severe infection. However, as we did not sample cells from the blood, we were not able to assess this part of the specific immune response. Our results are insofar not generalizable, as the cohort consisted of a well-defined group of young and healthy HCW with an increased exposure to SARS-CoV-2.

## Conclusions

In this real-life study using large-scale serology data, we evaluated SARS-CoV-2 infections during the Delta and Omicron waves in Switzerland and observed that hybrid immunity in HCW —compared to other immune status—was associated with the lowest risk of (re)infection and less symptoms in case of infection. Booster vaccination was associated with a risk reduction and with fewer symptoms of SARS-CoV-2 breakthrough infection, although this benefit seemed to fade during the Omicron period. Thus, our findings might inform healthcare providers and public health authorities in prioritizing SARS-CoV-2 vaccinations.

## Supporting information

**S1 STROBE Checklist. STROBE checklist.**
(PDF)

**S1 Methods. "Missing value imputation for covariates in multivariable models" and "Verification of proportional-hazard assumption in Cox regression."**
(PDF)

**S1 Table. Covariable definitions, levels, and time points when variables were obtained.**
(PDF)

**S2 Table. Hazard ratios with 95% confidence intervals from separate univariable Cox regression models for each predictor involved in the multivariable models regarding COVID-19 risk by period (Table 1 in main text).**
(PDF)

**S3 Table. Sensitivity analyses: Comparison of hazard ratios obtained in Cox models with three independent missing value imputations, without missing value imputation (i.e., complete case analysis), and with exclusion of events occurring during the period of variant overlap between December 6, 2021 and January 3, 2022.**
(PDF)

**S4 Table. Model with time-dependent impact of booster in the Omicron-dominant period.**
(PDF)

**S5 Table. Adjusted hazard ratios (HR) with 95% confidence intervals (CI) from multivariable Cox regression regarding risk of SARS-CoV-2 (re)infection; group N (no immunity) excluded and group V (vaccinated) defined as reference group.** Model additionally includes time from preimmunization to serology (i.e., months since last infection or vaccination) compared to the main analysis.
(PDF)

**S6 Table. Hazard ratios (HR) with 95% confidence intervals (CI) from multivariable Cox regression regarding risk of SARS-CoV-2 infection in the subgroup of those vaccinated but not infected (group V).** Model includes type of vaccine and timing of vaccinations between dose 1 and dose 2.
(PDF)

**S7 Table. Rate ratio (RR) and 95% confidence intervals (CI) from multivariable Poisson regression regarding number of symptoms reported from SARS-CoV-2 infections during the Delta and Omicron period.** Model includes only infections not preceded by booster or first vaccination.
(PDF)

**S8 Table. Rate ratio (RR) and 95% confidence intervals (CI) from multivariable Poisson regression regarding number of symptoms reported from SARS-CoV-2 infections during the Omicron period.** Model includes booster vaccine and is therefore restricted to groups V and H.
(PDF)

**S1 Fig. Definition of four groups by immune status and outcomes.** Created with BioRender.com.
(TIFF)

**S2 Fig. Flow sheet of participants in the SURPRISE study showing reasons (and respective number of participants) for exclusion from current analysis as well as participants within each immune status including number of subsequently vaccinated individuals, respectively.**
(PDF)

**S3 Fig. Time course of subsequent vaccinations (booster and new vaccinations).** N (no immunity): no reported infection and anti-N/-S negative and no previous SARS-CoV-2 vaccination; V (vaccinated): no reported infection and anti-N negative, but twice vaccinated; I (infected): infection reported or anti-N positive (at any time), but no vaccination; H (hybrid immunity): reported infection or anti-N positive (at any time) and vaccination ($\geq$1 dose).
(TIFF)

## Acknowledgments

The members of the SURPRISE study team are (in alphabetical order): Ulrike Besold, Angela Brucher, Thomas Egger, Andrée Friedl, Fabian Grässli, Sabine Güsewell, Eva Lemmenmeier, Christian R. Kahlert, Joelle Keller, Dorette Meier Kleeb, Philipp Kohler, Stefan P. Kuster, Onicio Leal, Dorette Meier Kleeb, Allison McGeer, J. Carsten Möller, Maja F. Müller, Vaxhid Musa, Manuela Ortner, Philip Rieder, Lorenz Risch, Markus Ruetti, Matthias Schlegel, Hans-Ruedi Schmid, Reto Stocker, Pietro Vernazza, Matthias von Kietzell, Danielle Vuichard-Gysin, and Benedikt Wiggli.

We would like to thank the employees of the participating healthcare institutions who either took part in this study themselves or supported it. Furthermore, we thank the laboratory staff for shipment, handling, and analysis of the blood samples.

## Author Contributions

**Conceptualization:** Sabine Güsewell, Allison McGeer, Lorenz Risch, Matthias Schlegel, Stefan P. Kuster, Christian R. Kahlert, Philipp Kohler.

**Data curation:** Sabine Güsewell, Thomas Egger, Onicio Leal, Angela Brucher, Eva Lemmenmeier, Dorette Meier Kleeb, J. Carsten Möller, Philip Rieder, Markus Rütti, Hans-Ruedi Schmid, Reto Stocker, Danielle Vuichard-Gysin, Benedikt Wiggli, Ulrike Besold, Andrée Friedl, Christian R. Kahlert, Philipp Kohler.

**Formal analysis:** Sabine Güsewell, Christian R. Kahlert, Philipp Kohler.

**Funding acquisition:** Christian R. Kahlert, Philipp Kohler.

**Investigation:** Angela Brucher, Eva Lemmenmeier, Dorette Meier Kleeb, J. Carsten Möller, Philip Rieder, Markus Rütti, Hans-Ruedi Schmid, Reto Stocker, Danielle Vuichard-Gysin, Benedikt Wiggli, Ulrike Besold, Andrée Friedl, Christian R. Kahlert, Philipp Kohler.

**Methodology:** Sabine Güsewell, Allison McGeer, Lorenz Risch, Matthias Schlegel, Stefan P. Kuster, Christian R. Kahlert, Philipp Kohler.

**Project administration:** Thomas Egger, Christian R. Kahlert, Philipp Kohler.

**Resources:** Christian R. Kahlert, Philipp Kohler.

**Software:** Sabine Güsewell, Thomas Egger, Onicio Leal, Christian R. Kahlert, Philipp Kohler.

**Supervision:** Allison McGeer, Matthias Schlegel, Stefan P. Kuster, Christian R. Kahlert, Philipp Kohler.

**Validation:** Sabine Güsewell, Onicio Leal, Angela Brucher, Eva Lemmenmeier, Dorette Meier Kleeb, J. Carsten Möller, Philip Rieder, Markus Rütti, Hans-Ruedi Schmid, Reto Stocker, Danielle Vuichard-Gysin, Benedikt Wiggli, Ulrike Besold, Lorenz Risch, Andrée Friedl, Christian R. Kahlert, Philipp Kohler.

**Visualization:** Sabine Güsewell, Christian R. Kahlert, Philipp Kohler.

**Writing – original draft:** Baharak Babouee Flury, Sabine Güsewell, Christian R. Kahlert, Philipp Kohler.

**Writing – review & editing:** Baharak Babouee Flury, Sabine Güsewell, Thomas Egger, Onicio Leal, Angela Brucher, Eva Lemmenmeier, Dorette Meier Kleeb, J. Carsten Möller, Philip Rieder, Markus Rütti, Hans-Ruedi Schmid, Reto Stocker, Danielle Vuichard-Gysin, Benedikt Wiggli, Ulrike Besold, Allison McGeer, Lorenz Risch, Andrée Friedl, Matthias Schlegel, Stefan P. Kuster, Christian R. Kahlert, Philipp Kohler.

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
