## [Editor Report · Decision Letter 0]

8 Jun 2022

Dear Dr Kahlert, 

Thank you for submitting your manuscript entitled "Risk and symptoms of COVID-19 during the Delta and Omicron waves according to baseline immune status and booster vaccination – a prospective multicentre cohort of health professionals in Switzerland" for consideration by PLOS Medicine.

Your manuscript has now been evaluated by the PLOS Medicine editorial staff and I am writing to let you know that we would like to send your submission out for external peer review.

Please re-submit your manuscript within two working days, i.e. by Jun 10 2022 11:59PM.

Kind regards,

Callam Davidson

Associate Editor

PLOS Medicine

---

## [Decision Letter · Decision Letter 1]

9 Aug 2022

Dear Dr. Kahlert,

Thank you very much for submitting your manuscript "Risk and symptoms of COVID-19 during the Delta and Omicron waves according to baseline immune status and booster vaccination – a prospective multicentre cohort of health professionals in Switzerland" (PMEDICINE-D-22-01834R1) for consideration at PLOS Medicine. 

Your paper was evaluated by an associate editor and discussed among all the editors here. It was also discussed with an academic editor with relevant expertise, and sent to independent reviewers, including a statistical reviewer. The reviews are appended at the bottom of this email and any accompanying reviewer attachments can be seen via the link below:

[LINK]

In light of these reviews, I am afraid that we will not be able to accept the manuscript for publication in the journal in its current form, but we would like to consider a revised version that addresses the reviewers' and editors' comments. Obviously we cannot make any decision about publication until we have seen the revised manuscript and your response, and we plan to seek re-review by one or more of the reviewers. 

We hope to receive your revised manuscript by Aug 30 2022 11:59PM. Please email us (plosmedicine@plos.org) if you have any questions or concerns.

We look forward to receiving your revised manuscript. 

Sincerely,

Callam Davidson, 

PLOS Medicine

plosmedicine.org

In the last sentence of the Abstract Methods and Findings section, please describe the main limitation(s) of the study's methodology."

Your study is observational and therefore causality cannot be inferred. Please remove language that implies causality, such as "Hybrid immunity and booster vaccination reduced the risk and symptom number of

SARS-CoV-2 infection during Delta- and Omicron-dominant periods" (Abstract Conclusions). Refer to associations instead. Other examples include use of the term 'impact' in the final paragraph of the Introduction.

As noted by Reviewer #3, your Introduction does not adequately describe prior work performed in this area. Please expand your Introduction to reflect the literature more accurately. 

Given that there are now multiple omicron subvariants in circulation, it may be useful to specify which omicron subvariant was predominant during the study period. 

Consider including an additional figure to clarify the groups included in your study, as the text description can at times be difficult to follow and a figure would help the reader visualise the design. 

Please include the completed STROBE checklist as Supporting Information. Please add the following statement, or similar, to the Methods: "This study is reported as per the Strengthening the Reporting of Observational Studies in Epidemiology (STROBE) guideline (S1 Checklist)."

Did your study have a prospective protocol or analysis plan? Please state this (either way) early in the Methods section.

Table S2 ought to be moved to the main text as it is central to the interpretation of the findings.

When reporting adjusted hazard ratios, please label these as such (aHR). 

When reporting significant findings, please quantify with both 95% CI and p-values where available.

Table 1: Definitions for abbreviations do not require an accompanying flag in the table itself. 

For the adjusted analyses presented in Table 1, please also present the crude analyses (unadjusted HRs) in the Supporting Information.

Line 245: Please update 'not anymore' to 'no longer'.

Line 251 and line 308: Please delete 'marginally' and include the p-value.

Throughout, please ensure p-values/HRs/95% CIs are reported to the same number of decimal places between the main text and the Tables/Figures.

In all Tables and Figures presenting adjusted analyses, please ensure you include the variables adjusted for in the respective legends. 

Line 280 and line 314: Please temper the use of causal language here, ('we show' could be updated to 'our observational findings suggest', or similar).

Line 289: Typo in 124,500.

Line 312: Your study was not designed to assess true vaccine effectiveness, please remove this.

The above noted concerns regarding causal language are also applicable to your Conclusions paragraph.

Please remove details of funding, author contributions, and conflicts of interest from the Acknowledgements. This information is captured via the submission form and, in the event of acceptance, will be published as metadata.

Comments from the reviewers:

Reviewer #1: Alex McConnachie, Statistical Review

This review looks at the use of statistics in the paper by Babouee Flury et al. The paper looks at COVID-19 infection/reinfection in relation to past infection/vaccination status, in a cohort of health workers, during the Delta and Omicron waves in Switzerland. Survival analysis methods (Kaplan-Meier, Cox regression) are used.

In general, the statistical aspects of the paper are very good. I have only a few observations.

Booster vaccination is included in the models as a time-varying covariate, but individuals are censored when they are vaccinated for the first time. Could vaccination status not also be seen as a time-varying covariate in these models? Similarly, if (re)infected, could that be seen as a change of status? This would provide additional follow-up time for these individuals. I am not suggesting this as the primary analysis, but such a model could be used as a sensitivity analysis, unless there is a good reason not to do it.

The adjusted regression models are based on individuals with complete baseline data only. Would multiple imputation of this missing data have been a better approach?

The analysis is repeated after exclusion of the N group. I am not sure what this achieves. Simply changing the reference group for the analysis to the V group would have done much the same thing. Looking at the coefficients in the different models, my guess is the results would be largely the same.

As ever, when fitting Cox models, there is always the question of the proportional hazards assumption. How was this checked? Was the assumption acceptable?

The tables report HRs to two decimal places, but the main text only uses one. I guess this is an editorial decision, but I prefer two decimal places.

I spotted a typo on line 260 - the point estimate of the HR for having a comorbidity at baseline is not within the confidence interval.

The authors fall into the trap of using causal language when describing an observational study (e.g. line 183 - "effect"; line 194 - "impact"). The associations may well be causal, but care needs to be taken to not overstate the results.

Reviewer #2: The authors are addressing an important question on the role of immune status and risk of SARS-CoV-2 infection and related symptoms. I appreciate the authors approach of observing multiple groups of various immune status as it is very interesting and needed information. 

The study design is large prospective cohort of health care professionals that were followed from September 2021 until the emergence of the Delta and Omicron variants, March 2022. This designed provides a valuable opportunity to assess infection as an outcome.

Title: Mentions severity of COVID-19 but unclear if severity of disease is reported. Please comment on any medically attended infection, hospitalization and deaths in this cohort. Please describe symptoms in the results section or supplement.

Below are a short list of data collection and definitions reported by the authors:

Weekly Questionnaires

Monthly follow up surveys

Self-reported exposures, NP swab results, symptoms associated with positive swabs, vaccinations

Periodic serology measurements

Group N = no infection, negative serology (anti N/S), no vaccination + first series data if applicable

Group V = no infection, negative serology (anti N), twice vaccinated + booster data if applicable

Group I = infection or anti N positive at any time, no vaccination + first series data if applicable

Group H = infection or anti N positive at any time, received greater than or equal to 1-dose + booster data if applicable

Exclusion criteria

Single vaccine dose + no previous infection

1st or 2nd dose between baseline serology and Sept 20th

Line-item questions:

Line 134: If no sequencing of variants were performed, how was a positive test attributed to a specific variant? 

Line 134: Self-reported NPS results were verified by the authors. Was this the same case for rapid antigen test at institutions that used this type of test? 

Line 142: Please clarify "twice vaccinated". Does this mean received 2-dose primary series within recommended timeframe? received 2 doses at anytime during the study period? etc.

Lines 148-149: Please clarify "single vaccine dose". It appears that Group H included at least those with 1-dose. Assuming single vaccine dose refers to vaccines that only have 1-dose but would like author's clarification. 

Line 149: When was baseline serology collected? The authors mentioned periodic serology measurements earlier in the text. Please briefly include when the measurement times occurred. Due to several mentions of dates, it may be helpful to write out the full dates i.e. September 20th, 2021 instead of September 20th. 

Lines 156-158: During variant overlap period, how were variant assignment decided for infections that occurred during circulations of both variants? I am concerned about that misclassification could impact the estimates. Please clarify how this was handled in the analysis. In Line 119, participants were recruited from Northern region as well. However, sequencing data was from Eastern region. Please include reasoning behind this method.

Lines 215-217: Any significant difference between groups V, I, and H?

Line 240: How were household infections data verified? Did all household contacts get tested if exposed or only if symptomatic? Please clarify in methods. 

Figure 1: Great figure. Consider including Northern Switzerland since there were participants recruited from that region. Possibly describe dates for booster vaccination and follow-up in the description.

Figure 2: Great figure. Consider including adjustments made in the description.

Supplement: If applicable, please include description of testing practices, serology data collection, laboratory procedures, and questionnaires.

Reviewer #3: The goal of this paper is to estimate the protective effects of vaccination and prior infection against SARS-CoV-2 infection and symptoms, predominantly during the Omicron variant wave of infection. The authors state that "It remains currently unknown in how far natural and/or vaccine-induced immunity protect against symptomatic infection by the Omicron variant, and how this can be further mitigated by booster vaccinations" (Introduction line 101). However, in the discussion the authors go on to cite the following 5 papers, each of which report on exactly that:

1. Effects of Previous Infection and Vaccination on Symptomatic Omicron Infections - Altarawneh et al. NEMJ 2022

2. Duration of mRNA vaccine protection against SARS-CoV-2 Omicron BA.1 and BA.2 subvariants in Qatar Chemaitelly et al. Nature Communications 2022

3. Protection with a Third Dose of mRNA Vaccine against SARS-CoV-2 Variants in Frontline Workers - Yoon et al. NEMJ 2022

4. Association Between 3 Doses of mRNA COVID-19 Vaccine and Symptomatic Infection Caused by the SARS-CoV-2 Omicron and Delta Variants - Accorsi et al. JAMA 2022

5. Covid-19 Vaccine Effectiveness against the Omicron (B.1.1.529) Variant - Andrews et al. NEMJ 2022

There are also at least a further 4 papers that they don't cite, which add further evidence:

1. Observed protection against SARS-CoV-2 reinfection following a primary infection: A Danish cohort study among unvaccinated using two years of nationwide PCR-test data - Michlmayr et al. The Lancet Regional Health Europe 2022

2. SARS-CoV-2 Omicron Symptomatic Infections in Previously Infected or Vaccinated South African Healthcare Workers - Nunes et al. Vaccines 2022

3. Effectiveness of BNT162b2 Vaccine against Omicron in Children 5 to 11 Years of Age - Tan et al. NEMJ 2022

4. Effectiveness of a fourth dose of covid-19 mRNA vaccine against the omicron variant among long term care residents in Ontario, Canada: test negative design study - Grewal et al. BMJ 2022

Similarly, there are many papers assessing protection against Delta, which was dominant at the beginning of the study period in this paper.

Overall, there's a substantial body of existing evidence on the protection against SARS-CoV-2 Delta and Omicron variants associated with prior infection and vaccination. The authors need to acknowledge this and clearly situate their study within the context of the existing literature. As a relatively small occupational cohort, it isn't clear to me that this study makes a substantial contribution to the literature, so it doesn't seem like a good fit for PLoS Medicine.

[LINK]

---

## [Decision Letter · Decision Letter 2]

3 Oct 2022

Dear Dr. Kahlert,

Thank you very much for re-submitting your manuscript "Risk and symptoms of COVID-19 during the Delta and Omicron waves according to baseline immune status and booster vaccination – a prospective multicentre cohort of health professionals in Switzerland" (PMEDICINE-D-22-01834R2) for review by PLOS Medicine.

I have discussed the paper with my colleagues and the academic editor and it was also seen again by two reviewers. I am pleased to say that provided the remaining editorial and production issues are dealt with we are planning to accept the paper for publication in the journal.

[LINK]

We look forward to receiving the revised manuscript by Oct 10 2022 11:59PM.   

Sincerely,

Callam Davidson, 

Associate Editor 

PLOS Medicine

plosmedicine.org

Requests from Editors:

Please ensure that the additional comments from Reviewer #1 are addressed. 

Please update your title to “Risk and symptoms of COVID-19 in health professionals according to baseline immune status and booster vaccination during the Delta and Omicron waves in Switzerland: a multicentre cohort study”.

PLOS Medicine requires that the de-identified data underlying the specific results in a published article be made available, without restrictions on access, in a public repository or as Supporting Information at the time of article publication, provided it is legal and ethical to do so. Please see the policy at 

http://journals.plos.org/plosmedicine/s/data-availability

and FAQs at 

http://journals.plos.org/plosmedicine/s/data-availability#loc-faqs-for-data-policy

The Data Availability Statement (DAS) requires revision. For each data source used in your study: 

If the data are freely available upon request, please note this and state the owner of the data set and contact information for data requests (web or email address). Note that a study author cannot be the contact person for the data.

As noted in your response to Reviewer #2, comment 3 (R2.3), please include the additional information in your Methods (around lines 181-182). 

It was noted that author OL is affiliated with Epitrack – please confirm the involvement of Epitrack and provide comment on whether you feel this affiliation ought to be declared as a competing interest.

Please shorten the bullet point at lines 90-94 (single sentence bullets are preferred in the Author Summary).

Please include the number of participants in your Author Summary (this can be added to the bullet at lines 99-101). 

Line 150: ‘…we aimed to determine...’

Line 161: Please include details of how informed consent was sought from participants and specify whether this was verbal or written. 

Thank you for creating Figure S1 in response to my previous comment. As the figure was produced using BioRender, please confirm that the usage rights for this figure are compatible with PLOS Medicine’s Creative Commons Attribution (CC BY) license (which is applied to all figures we publish). The following may be useful: https://help.biorender.com/en/articles/5898558-where-can-i-use-my-illustrations

Lines 225 and 259: To ensure correct hyperlinking, please ensure references to Supporting Information are labelled as described here: https://journals.plos.org/plosmedicine/s/supporting-information (i.e., S1 Methods, S1 Checklist, etc.). 

Please define the groups in the legend of Figure S2.

Table 2: Please move the reference to Table S2 from the Table 2 title to the legend (likewise for the corresponding reference in Table S2).

Lines 329 and 420: Given the observational design, please refer to associations rather than effects.

Please relocate the discussion of strengths at lines 330-333 to later in the Discussion (where strengths and limitations are discussed). As far as is possible, the Discussion should be organised as follows: a short, clear summary of the article's findings; what the study adds to existing research and where and why the results may differ from previous research; strengths and limitations of the study; implications and next steps for research, clinical practice, and/or public policy; one-paragraph conclusion.

Comments from Reviewers:

Reviewer #1: Alex McConnachie, Statistical Review

I thank the authors for their consideration of my original points, and I am generally happy with their responses.

However, whilst Table S3 shows that the model results from three imputed datasets are similar (as would be expected), they are not identical (or nearly identical, as the authors suggest). This table shows that there is some variation between imputations. By using a single imputed dataset, the analysis fails to take account of the added uncertainty in the results caused by the fact that some data are missing. By using multiple imputation, this uncertainty can be represented through the confidence intervals around the model parameters.

Three is a small number of imputed datasets. Personally I use ten, though others would argue for more. As far as I know, there is no penalty to doing extra imputations. The important thing is to pool the results from the models fitted to each dataset. This is built in to the mice package in R, so should be straightforward.

Reviewer #3: The authors have now addressed all my concerns from the first version of the manuscript. The revised version situates the study much more clearly in the existing literature - it is now clear what evidence already exists, and what this study adds.

[LINK]

---

## [Editor Report · Decision Letter 3]

14 Oct 2022

Dear Dr Kahlert, 

On behalf of my colleagues and the Academic Editor, Professor James Beeson, I am pleased to inform you that we have agreed to publish your manuscript "Risk and symptoms of COVID-19 in health professionals according to baseline immune status and booster vaccination during the Delta and Omicron waves in Switzerland – a multicentre cohort study" (PMEDICINE-D-22-01834R3) in PLOS Medicine.

Thank you for you considerate responses to editor and reviewer comments. To help us extend the reach of your research, please provide any Twitter handle(s) that would be appropriate to tag, including your own, your coauthors’, your institution, funder, or lab.

PRESS

Sincerely, 

Philippa Dodd MBBS MRCP PhD

Associate Editor

PLOS Medicine

On behalf of - Callam Davidson 

Associate Editor 

PLOS Medicine